# Compensation of Measurement Uncertainty in a Remote Fetal Monitor

**Sadot Arciniega-Montiel [1], Guillermo Ronquillo-Lomeli [1,2,*] , Roberto Salas-Zúñiga [3],**
**Tomás Salgado-Jiménez [1] and Leonardo Barriga-Rodríguez [1]**

[1]  Department of Energy, Center for Engineering and Industrial Development, Santiago de Queretaro 76125, Mexico; sadot@cidesi.edu.mx (S.A.-M.); tsalgado@cidesi.edu.mx (T.S.-J.); lbarriga@cidesi.edu.mx (L.B.-R.)
[2]  Faculty of Engineering, Autonomous University of Queretaro, Santiago de Queretaro 76010, Mexico
[3]  Engineering and Technology Center, Santiago de Queretaro 76079, Mexico; roberto.salas.correo@gmail.com
[*]  Correspondence: gronquillo@cidesi.edu.mx; Tel.: +52-442-211-9800

**Abstract:** The perinatal mortality rate is very high throughout the world. A fetal monitor may be used remotely, and this would tackle the problem of continuous monitoring of high-risk pregnancies. There is evidence that current technology is of low reliability, and, therefore, of low precision to identify fetal health. In medical technological implementation, a safe, efficient, and reliable operation must be guaranteed, and the main problem is that remote fetal monitor gathers just a few samples, so the hypothesis of classical theory is not met. We are proposing an approach that improves the data's lack of reliability that accompanies the use of a remote fetal monitor. The method refers to how, by using the existing technologies and the initial experimental data, it is possible to apply probabilistic models that are truly representative of each application. This leads to the characterization of properties of the statistics used to generate a representative probabilistic model without the need to consider the hard suppositions. Results show that, for every case study, it was possible to improve estimations of measurement uncertainty. The proposed method is a useful tool to increase the reliability of medical technology, especially for pieces of equipment where a health care professional is not available.

**Keywords:** remote fetal monitor; measurement uncertainty; standard deviation; Monte-Carlo method (MMC); efficient estimator

## 1. Introduction

The perinatal mortality rate is very high throughout the world. According to the World Health Organization (WHO), it is estimated that 800 women die every day due to pregnancy or birth-related causes. From all pregnancies, four out of ten women are at high risk. There is no hospital infrastructure for the timely diagnosis of fetal distress due to overpopulation.

Continuous remote monitoring technology would provide a solution to solve some problems in high-risk pregnancies. Patients would no longer need to be at the hospital, and, from the comfort of their homes, would directly carry out and forward the studies prescribed by their physicians. The monitoring equipment could be configured for providing the corresponding studies directly by the patient without the need of a specialist. Patients may feel calm and reassured, knowing that their studies, once transmitted by the monitor, will be checked and assessed by a specialist.

Fetal monitoring technology faces reliability issues. When compared against intermittent auscultation, the use of technology reduces perinatal mortality [1–3]. However, some other studies from the 1970s and 1980s did not show similar results [4], and the contribution of technology to reduce perinatal mortality, as observed in the last decades, is still under discussion. On the other hand, the

increase in rates of births via Caesarian section has become a matter of public interest in many countries, and there is a general belief that continuous monitoring during childbirth (CTG) has contributed to this.

CTG is subject to a poor observer interpretation, particularly while assessing variability and de-acceleration, in addition to the classification of tracing [5–7]. Diagnosis depends on the criteria used for tracing analysis [8], so objective guidelines are required to provide a practical approach.

It is necessary to have an approach that improves the data reliability produced by the remote fetal monitor to increase the diagnosis precision [9,10]. Fetal monitoring systems should yield reliable data, regardless of the conditions of use, such as location, installation, number of samples, etc., for the revision of fetal health and the prompt response of healthcare professionals.

In medical technology implementation, a safe and efficient operation must be guaranteed [11]. Attention is focused on the practical implementation of technology within a sanitary environment. The simple fact of having the necessary technology is not a whole solution. The processes of planning, evaluation, selection, and implementation of new technologies, all in concept and context, must be considered. This may imply a direct solution to specific assessments of sanitary technology. When developing new technology, guaranteeing precision and reliability for the chosen environment contributes to achieving safer results and sound medical attention.

Under these conditions, the main problem is to reassure the reliability of the data obtained by the remote fetal monitor, since health-care professionals would not be present, and there would be only a few samples of variables (heart rate (HR), uterine activity) that lead to the detection of anomalies. Therefore, the analysis of measurement uncertainty becomes necessary.

For reliable estimation of measurement uncertainty, it is necessary to guarantee some properties (distribution independence, a large sample size, and a normal distribution). Health-care professionals are not familiar with the measurement process carried out by remote fetal monitoring, and, in general, only have a few samples. This means it is not possible to verify the hypotheses required so that the evaluator may have the properties required by a good estimator. By means of a statistical approach, several works related to uncertainty estimation have been conducted, for cases where the hypothesis of classical theory is not verified.

A fetal monitor is a stochastic process that has few samples available. Therefore, it is very difficult to model by using conventional methods. Different approaches have been used when the hypothesis of the classical theory is not verified, including the Monte-Carlo method (MCM) [12], parametric and non-parametric theory formulation [13], adaptive estimators [14], spectrum-based estimation [15], stochastic gradient [16], and mean square error [17] estimators with Cramer–Rao type restriction and reliability estimation under measures degradation [18].

The MCM is a widely used tool for modeling stochastic systems since with a large enough sample, the probability that an estimator will deviate from the expected value can be as small as required. The MCM is used to get information related to estimator properties. The parameters of softening and form of estimating function are adjusted to modify the performance of maximum probability estimators. The MCM can be used for modeling unknown distribution functions. The parametric and non-parametric models are wide probability function estimation methods; theory formulation has been developed considering local asymptotic length and restrictions in sample size. Adaptive estimators that guarantee the consistency in finite samples for any distribution function are helpful in the fetal monitor application. The term adaptive integrates the concept that such estimators adapt to the sample using data to estimate a non-parametric density function. Adaptive estimators have been applied using the least-squares method with finite samples, reaching substantial increases in efficiency, but with hypotheses that are difficult to verify for applications with a few samples. The spectrum-based method is useful when the relationship between observations and parameter model is noisy, potentially non-linear, and not invertible. The stochastic gradient estimation is a numeric method that is stable and robust for fitting parameters and predicting standard errors well. Even though only the typical case, with some modifications, was considered, the hypothesis that let the Cramer–Rao type limit such estimators have been changed. The mean square error can be improved by implementing estimators

with parametric restrictions that do not satisfy certain conditions. With a new definition of bias in restricted surroundings, a Cramer-Rao restriction type was determined for the pondered mean square error estimators, for normal distribution only. Correlated perturbations were not considered. The reliability estimation for a system whose degradation measures are monotonous, can obtain the fault time distribution, and, therefore, system reliability.

The remote fetal monitor has a context application. The aim of this work is to obtain an integral maternal-fetal watch program, integrated by a technological platform that allows the timely detection of perturbations in baby heart rate, that enable us to remotely send information to an obstetric monitoring center for timely and reliable diagnosis of the patient.

When modeling a process, the most complicated component of the model is represented in probabilistic terms, and the hypotheses taken as valid in many situations are not met, and the results are not valid. This work presents a proposal of how, using the existing theory, with a deterministic model and the Monte-Carlo method [12], it is possible to apply probabilistic models that are genuinely representative of the application, and that characterize the statistics' properties used. In a specific application, the statistical properties of interest are generally mean, variance, extreme values, dependence on sample size, bias, etc. [13–17]. In general, only the mean and variance parameters are considered to obtain a probabilistic model because they characterize the uniform distributions that represent the worst-case scenario (total ignorance of the model) and the Gaussian distributions considered as typical or normal. However, to estimate these parameters, it is necessary to carry out a sampling to obtain more information about the system model. This additional information is not used in practice. This work proposes a new approach to include experimental data and to generate a representative probabilistic model without considering difficult suppositions. The suggested procedure is applied to a fetal monitor to correct the standard deviation estimation considering the noise type that occurs during the regular operation of the device.

## 2. Uncertainty Estimation Model

One of the main problems in metrology is the estimation of measurement uncertainty. According to [19], "estimated standard deviation associated to output, or measurement estimation is called combined standard deviation, and it is determined from the estimated standard deviation associated for each estimation of output, called standard uncertainty". In general, the process to be measured is ignored, and regularly, there are only just a few samples available. Measurements have defects that lead to an error in the result of such measurements. Although it is not possible to compensate for the random error in a measurement result, its effect can generally be reduced by increasing the number of observations until the expectations or expected value becomes zero.

Modeling of stochastic systems has been thoroughly studied, considering the hypothesis of the great numbers law. However, a measurement system is a stochastic system, with a limited or finite number of samples. The remote fetal monitor is a stochastic measurement system with a limited number of samples. Below an uncertainty estimation model for a measurement system with limited samples is presented.

A probability space $(\Omega, \mathcal{F}_0, P)$ for a random variable $X : \Omega \to \mathbb{R}^n$ where the conditional expectation $E\{(\cdot), \mathcal{F}_0\}$ concerning to sigma-algebra $\mathcal{F}_0 = \left\{ \phi, \ \Omega \right\}$ [20] may be expressed as

$$E\{X|\mathcal{F}_0\} = E\{X\} \tag{1}$$

This means that the best estimation for the random variable, when there is no a priori knowledge of the process, is the mathematical expectation, and the associated standard uncertainty ($u_c$) may be obtained from the probability density function (PDF) of the random variable [21]. Therefore, the mean is proposed to estimate the following sample, and standard uncertainty is obtained from the standard deviation estimator.

Let $A$ be the event where a device does not fail, the reliability is by definition $P(A)$ if the indicator function is considered $\chi_A$,

$$\chi_A(\omega) = \begin{cases} 1, & \omega \in A \\ 0, & \omega \notin A \end{cases} \tag{2}$$

This means that the problem of calculating reliability is changed for the problem of estimating the expected value, for which the use of the Monte-Carlo method in this work is necessary.

In the estimation of the mathematical expectation, the equation regularly used is

$$\overline{X}_n = \frac{1}{n} \sum_{k=1}^{n} X_k, \tag{3}$$

being $n$ the sample number.

In the normal case with independent samples, the mean estimator is optimal in the sense of the mean square error. In addition, the large number theorem is consistent for large samples, and its distribution is known by the central limit theorem [20], and is efficient [22].

In the general case, the estimator $\overline{X}_n$ is a random variable that depends on:

1. Sample size, $n$
2. Distribution, $F_X$
3. The degree of probabilistic dependence between samples, in addition to the fact that only linear or correlation dependence is considered $\rho_{X_i X_j}$.

To measure the quality of this, estimate the standard deviation is used (called uncertainty in the metrology area). This standard deviation is also estimated; this is done with $S_n$ defined as

$$S_n := \sqrt{S_n^2}, \tag{4}$$

$$S_n^2 := \frac{1}{n-1} \sum_{k=1}^{n} (x_k - \overline{x}_n)^2, \tag{5}$$

for independent samples with the same distribution we get,

$$E\left\{S_{\overline{X}_n}^2\right\} = \frac{E\left\{S_X^2\right\}}{n} = \frac{\sigma_X^2}{n}. \tag{6}$$

So, the problem of determining the quality of the estimate focuses on determining $\sigma_X$. However, we must identify the restrictions in the estimation.

A widespread practice is to consider that if

$$\left\{S_n^2\right\} = \sigma^2, \tag{7}$$

then,

$$E\{S_n\} = \sigma. \tag{8}$$

Nonetheless, defining by z a random variable,

$$z(\omega) = 1, \ \forall \ \omega \in \Omega, \tag{9}$$

and applying Cauchy–Bunyakowskii–Schwartz inequality [20],

$$E\{S_n\} = E\{S_n z\} \leq \sqrt{E\left\{S_n^2\right\}E\{z^2\}} = \sqrt{E\left\{S_n^2\right\}}, \tag{10}$$

that is

$$E\{S_n\} \leq \sigma, \tag{11}$$

this inequality depends on all these three: sample size $n$, distribution function $F_X$, and correlation dependence $\rho_{X_i X_j}$.

For a sample, $X = (X_1, \cdots, X_N)$ corresponds a density function $f_X(X, \theta)$ for which the score $V(\theta)$ is defined

$$V(\theta) \equiv V_X(\theta) \log f_X(X; \theta), \tag{12}$$

and the Fisher information matrix

$$I(\theta) E \left\{ \left[ \frac{\partial V(\theta)}{\partial \theta} \right]^2 \right\}. \tag{13}$$

The Cramer–Rao theorem is fulfilled [20] where if $\hat{\theta}_n = \hat{\theta}(X_1, \ldots, X_n)$ is an unbiased estimator of $\theta^\infty$ then

$$E_\theta \left\{ \| \hat{\theta}_n - \theta \|^2 \right\} \geq \frac{1}{n I(\theta)} \tag{14}$$

and also depend on sample size $n$, distribution $F_X$, and correlation dependence $\rho_{X_i X_j}$.

This means that in the case of the fetal monitor where there are few samples, it is possible to determine the boundaries for the reliability estimation and its uncertainty using the Monte-Carlo method.

## 3. Uncertainty Estimation Approach

This section describes a proposed approach to obtain a reliability estimator for heart rate measurement.

Reliability $R$, the probability that the measurement is correct, is a function of the measurement device, the perturbations represented by its distribution function and its correlation, in addition to the number of data used in the estimation.

To estimate the reliability and the uncertainty in the measurement, the following procedure is proposed:

1.  System model: device model and perturbation model.
2.  Distribution function estimation of the perturbations and correlation function.
3.  Estimators properties analysis using the Monte-Carlo method.
4.  Correction factors calculation to compensate estimators.
5.  Confidence intervals for the $R$ estimation.

### 3.1. System Model

In the block diagram of Figure 1, a system of single input $u_k$ and single output $y_k$ (SISO) is shown in simplified form. Noise $\xi_k$ presence is assumed a priori, and, in this case, white noise properties are attributed, which is presented as additive noise $r_k$ in the system output.

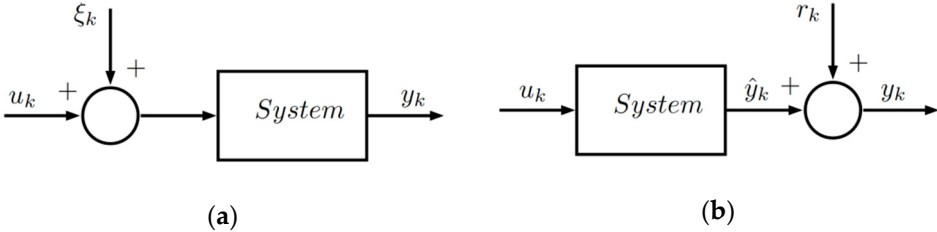

**(a)**                                             **(b)**

**Figure 1.** Single-input and single-output (SISO) system: (**a**) With white noise to the input; (**b**) With additive noise to the output.

It is considered that an input–output data set of the system is available $(y_k,\ u_k; k = 1,\ 2, \cdots, n)$ where $n$ is the measurement number taken from the input and output for modeling.

### 3.1.1. Device Model

To describe the dynamic behavior from a set of input and output measurements, an auto-regressive and exogenous (ARX) [23] (p. 81) linear model structure is assumed. A scheme for system modeling is shown in Figure 2.

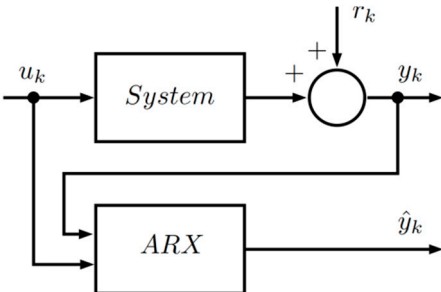

**Figure 2.** Scheme for linear system modeling based on auto-regressive and exogenous (ARX) structure.

The ARX structure model is

$$y_k = \phi_k^T \theta + \xi_k, \tag{15}$$

$\phi_k = [y_{k-1}, \cdots, y_{k-l}, u_k, \cdots, u_{k-l}]^T$ is the regression vector, $\theta = [a_1, \cdots, a_l, b_0, \cdots, b_l]^T$ is the parameter vector, while $l$ is the system degree. Based on Equation (15), the relationship between the input and the output in a compact way can be written in the following vector form.

$$Y = \Phi \theta + \Xi, \tag{16}$$

where $Y = [y_1, y_2, \cdots, y_n]^T$, $\Phi = \left[\Phi_1^T, \Phi_2^T, \cdots, \Phi_n^T\right]^T$, and $\Xi = [\xi_1, \xi_2, \cdots, \xi_n]^T$.

With this information and applying the least-squares method [24] (pp. 17–27), a deterministic model is obtained which represents the fetal monitor dynamic, the solution is

$$\hat{\theta} = \left[\Phi^T \Phi\right]^{-1} \Phi^T Y. \tag{17}$$

Finally, the relationship between white noise $\xi_k$ in the system input and additive noise $r_k$ in the system output for the ARX model structure is

$$R(z) = \frac{1}{1 - A(z)} \Xi(z), \tag{18}$$

where $A(z) = \hat{a}_1 z^{-1} + \cdots + \hat{a}_l z^{-l}$.

### 3.1.2. Perturbation Model

The probabilistic part is modeled by a stochastic process $\{\xi_k\}$, which must be strongly stationary, uncorrelated, with a cumulative distribution function (CDF) $F_{\Xi_k}$. These hypotheses are experimentally validated. The probabilistic model is used to determine the properties of the estimators used to determine the heart rate; considering that the noise $\xi_k$ is naturally present and that it is transformed into $r_k$ by the system during the route that the signal travels from the fetus heart through amniotic fluid, body fat, etc., to the fetal monitor output. Figure 3 shows a block diagram of the process for estimating the error distribution function $F_{\Xi_k}$.

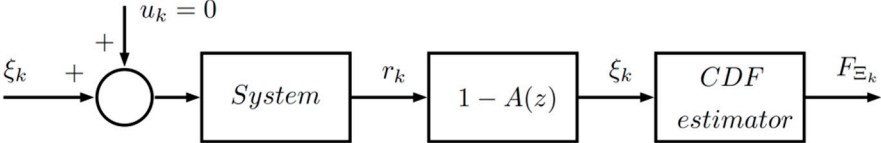

**Figure 3.** Estimation process of the error distribution function $F_{\Xi_k}$.

### 3.2. Distribution Function Estimation of the Perturbations and Correlation Function

For each patient, a stochastic model is determined, which describes the perturbation present in the fetal heart rate signal, which exists due to the individual anatomical characteristics of the patients, amniotic fluid, body fat, etc.

The stochastic model is obtained from data acquired by clinical studies using the patient's fetal heart rate monitor. Each clinical study (sample) is a set of measurements $(y_k, \ k = 1, \ 2, \cdots, n)$.

Using the data-normalized histogram, the probability density function (PDF) is estimated for $n$ samples $f_{\Xi_n}$, numerically integrating and using some interpolator an estimator is obtained for the probability distribution function $F_{\Xi_n}$ of the perturbations, which is used to build a $\xi$ simulator required in the Monte-Carlo method. Each sample that is added improves the probability distribution function estimation, so that, the more samples available, the better the estimator.

To guarantee that the bias of the expectation estimate is zero, the correlation between the perturbation and the regression vector used in the least-squares model must be verified as zero. For this, a correlation degree estimate is made [20,22] that exists between the perturbations and the measured signal. The correlation is defined as follows

$$\rho_{XY}(k) := \frac{E\left\{X_{j+k}Y_j\right\}}{\sqrt{E\left\{X_{j+k}^2\right\}E\left\{Y_j^2\right\}}}, \tag{19}$$

where $\rho_{XY}$ represents the correlation function between the random variables $X$ and $Y$. This magnitude is estimated using

$$\hat{\rho}_{XY}(k) = \frac{\frac{1}{n}\sum_{j=1}^{n}X_{j+k}Y_j}{\sqrt{\frac{1}{n}\sum_{j=1}^{n}X_{j+k}^2\frac{1}{n}\sum_{j=1}^{n}Y_j^2}}. \tag{20}$$

If there is a substantial correlation, a filter must be designed so that the perturbation behaves with white noise features.

### 3.3. Estimators Properties Analysis Using the Monte-Carlo Method

After obtaining the models of the system and the perturbations, it is necessary to estimate the value of the fetal heart rate. Simulated samples from the inverse distribution function $F_{\Xi_n}^{-1}F_-(\Xi\_n)$ were used for this estimation. To carry out this task, the sample mean $\overline{Y}_n$ and the sample deviation $S_n$ were used. The mean is useful for determining the fetal heart rate value and the standard deviation for quantifying the quality of the estimate, known as uncertainty in the metrological area.

However, it should be noted that for these estimators to be used properly, it is necessary to determine their properties specifically for each patient. The most important property to determine is how the length of sample $K$ impacts the distribution function.

On the other hand, the sample mean $\overline{Y}_n$ and the standard deviation $S_n$ are in turn random variables that are characterized by their probability distribution function $F_{\overline{Y}_n}$ and $F_{S_n}$, respectively. These functions in classical theory can be approximated with large samples and hypotheses of probabilistic independence, which turned out to be difficult to verify in this application. To solve this mathematical problem, the Monte-Carlo method is proposed as a tool to determine these distribution

functions without the use of these hypotheses. The procedure is presented in Figure 4, where $N$ is the number of samples used to simulate $\Xi_k$ and $K$ the number of samples to determine $F_{\overline{Y}_n}$ and $F_{S_n}$.

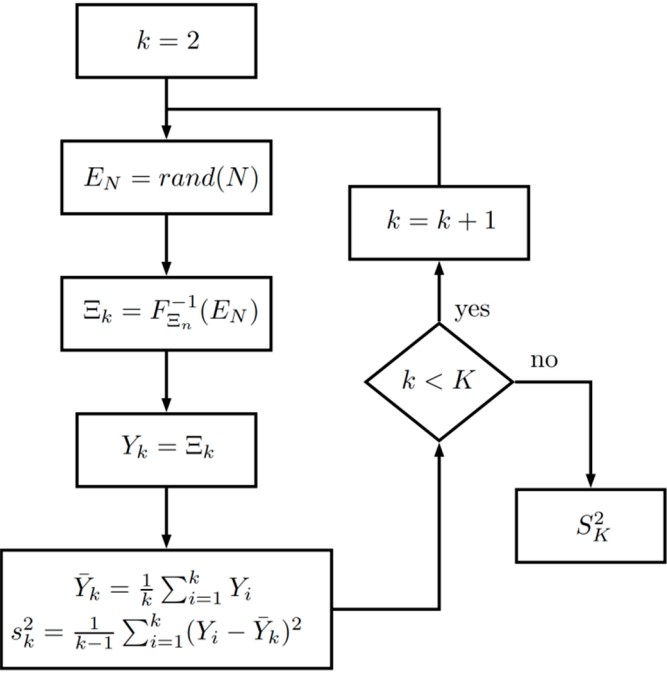

**Figure 4.** Simulation procedure with the Monte-Carlo method (MMC) to determine the variance $s_k^2$.

This procedure allows us to estimate the variance $\sigma^2$ because if $K$ is large, the following is fulfilled

$$E\{Y\} = E\{\overline{Y}_K\}, \tag{21}$$

$$S_K^2 = \frac{1}{K-1} \sum_{i=1}^{K} [Y_i - E\{Y\}]^2, \tag{22}$$

$$\sigma^2 = E\{S_K^2\}. \tag{23}$$

### 3.4. Correction Factors Calculation to Compensate Estimators

This procedure removes the constraints that exist to achieve an adequate estimate of reliability and provides the factors that must be included to eliminate the estimators' bias. These factors are

1. Additive factor for $\overline{Y}_k$ $\left(\left[\Phi^T\Phi\right]^{-1}\Phi^T\Xi\right)$
2. Multiplicative factor for $S_k$ $(C_k)$

In our case, the $\Phi$ and $\Xi$ turned out to be uncorrelated, so this additive term is close to zero and was not considered in this study.

The multiplicative factor for $S_k$ is obtained from the Cauchy–Bunyakowskii–Schwartz inequality described in Equation (10). In many cases where $E\{S_k\} < \sigma$, the question is how far is $E\{S_k\}$ from $\sigma$, so we will introduce a factor $c_k$ that eliminates this bias, that is to say

$$E\{c_k S_k\} = \sigma \tag{24}$$

The new standard deviation estimator $\sigma$ will be $c_k S_k$, this factor is estimated using the Monte-Carlo method and the relation

$$c_k = \frac{\sqrt{E\{(Y - E\{Y\})^2\}}}{\sqrt{\frac{1}{k-1}\sum_{i=1}^{k}\left(Y_i - \overline{Y}_k\right)^2}} = \frac{\sigma}{S_k}. \tag{25}$$

*3.5. Confidence Intervals for the R Estimation*

When clinical studies are performed on patients, new data are obtained, which must be analyzed to determine if they are correct. For carrying out this, confidence intervals are constructed on-line to ensure that the test is correct with a confidence level $\alpha$.

The last sample's variations are analyzed. These variations are measured by standard deviation (s), which is calculated by means of the $s_n$ estimator, considering the process defined by Y and the set of samples.

In a process defined by $Y$ with $N$ clinical studies set $Y_n$, $(n = 1, 2, \cdots, N)$, each clinical study has $M$ samples $[Y_n]_m$, $(m = 1, 2, \cdots, M)$. The variations are measured by the standard deviation $s_Y$, which is estimated through the corrected $s_n$.

From a statistical point of view, a measurement is considered incorrect if it is an atypical value. The criteria to decide is built by considering the probability that the current measurement will fall outside a range defined by a standard deviation estimator, considering the necessary compensations for the used estimators.

Now, we will determine a criterion to eliminate measurements that, with a high probability, may be wrong, since values will be far from most of the measurements, known as atypical data.

To do so, the Chebyshev approximation [25] is used

$$P\left(\frac{|Y - \mu_Y|}{\sigma_Y} \geq k\right) \leq \frac{1}{k^2}, \tag{26}$$

$$P\left(\frac{|Y - \mu_Y|}{\sigma_Y} < k\right) \geq 1 - \frac{1}{k^2}, \tag{27}$$

where $\mu_Y$ and $\sigma_Y$ are the mean and the standard deviation of the process, respectively.

If no further information is available but considering that new information is arriving every time a measurement is taken, the following statistics are considered

$$P\left(\frac{|Y - \mu_Y|}{c\, s_Y} < k\right) \geq 1 - \frac{1}{k^2}. \tag{28}$$

where $c$ is a correction factor due to the finite number of samples, and to the law of standard deviation spread?

Since $\mu_Y$, $s_Y$, and $c$ are typically unknown, these parameters are replaced by their estimators $\overline{y}_n$, $s_n$, and $c_n$, respectively. Finally, the selection criterion is obtained

$$P\left(\frac{|Y_n - \overline{y}_n|}{c_n s_n} < k\right) \geq 1 - \frac{1}{k^2}, \tag{29}$$

which is equivalent to

$$\left(Y_n - \overline{y}_n\right)^2 < \beta s_n^2, \tag{30}$$

where $\beta = k^2 c_n^2$, and $c_n$ is the correction factor that corresponds to the $n^{th}$ clinical study.

Figure 5 shows the flowchart of the reliability estimation algorithm based on Chebyshev's inequality.

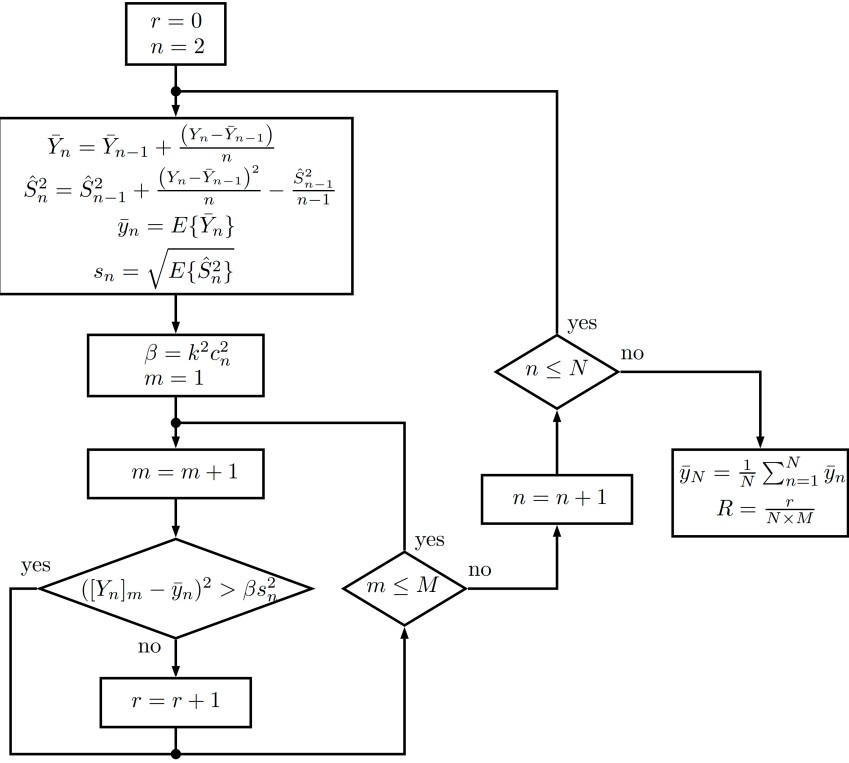

**Figure 5.** Reliability estimation algorithm.

### 3.6. Remote Fetal Monitoring Equipment

A fetal monitor was developed to provide a reliable remote monitoring of both mother and fetus during the pregnancy process. The fetal monitor has the capacity of recording fetal heart rate, uterine activity, and fetal movement. Its compact design allows the mother to use it herself. In addition, the fetal monitor allows the recording of studies, and features the capacity to send them directly to the obstetric monitoring center (OMC) to be evaluated by a specialist.

The purpose of this medical device is to provide monitoring of high-risk pregnancies, especially those in marginal areas, who cannot go to a clinical specialist periodically to reduce fetal distress, and, therefore, perinatal mortality in Mexico.

The fetal monitor is an embedded system, which includes a digital signal processor that allows for an adequate calculation to achieve higher precision in fetal heart rate reliability and signs of uterine activity. The algorithm developed included the proposed methodology of this work to guarantee the reliability of distance measurement. This instrument has enough hardware resources, such as remote connectivity by means of a mobile phone, through the general package radio service (GPRS), geo-referenced transmission, an alarm indicator for the patient, a speaker to listen to baby heartbeats, a flash memory to store studies and two Doppler ultrasound sensors, of 1.2 MHz in frequency, and 350 mW for output power, for the acquisition and processing of fetal monitoring variables. A block diagram with these functions is presented in Figure 6.

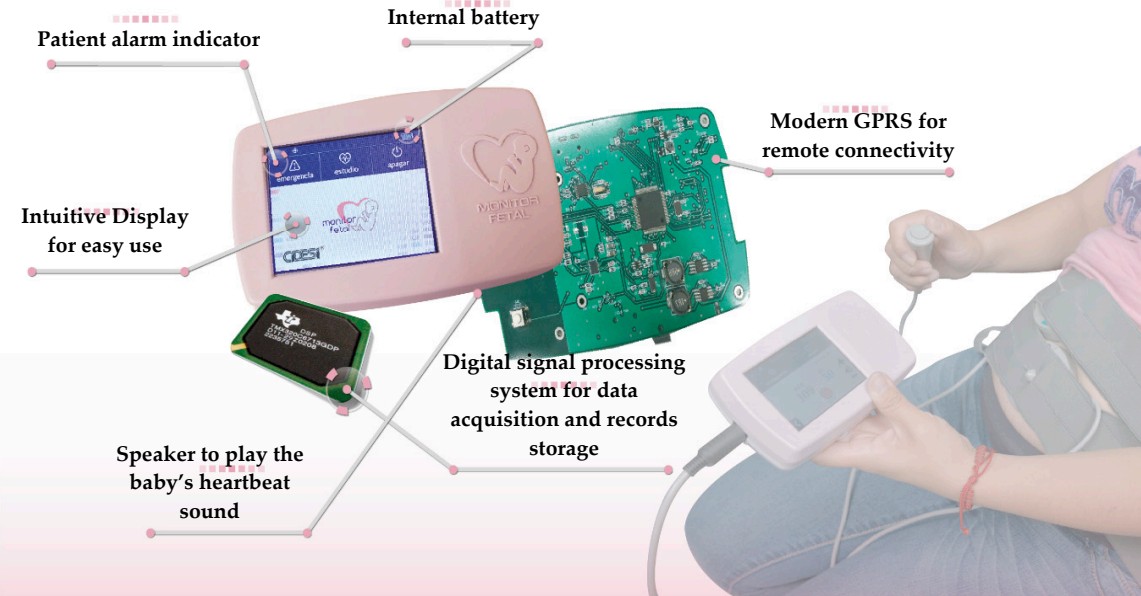

**Figure 6.** Scheme of the fetal monitor used for testing.

### 3.7. Experimental Clinical Studies

During experimental tests, several fetal monitoring instruments, in remote sites, were connected to patients and sent data signals to the OMC based at the Central Hospital. The patients were informed about their data usage and consented to publish their data for research purposes.

Data collected at OMC were primary signals without processing, that is, directly from the analog–digital converter of the instrument. For each test, a set of data with fetal heart rate and signs of uterine activity was stored. Table 1 shows the clinic studies in detail, to validate the proposed approach.

**Table 1.** Clinical Studies for approach validation.

| Patient Name | Age (Year) | Studies Number | Place | Distance from OMC |
| --- | --- | --- | --- | --- |
| Subject 1 | 37 | 42 | Toliman | 81.7 Km |
| Subject 2 | 41 | 45 | Jalpan de Serra | 187 Km |
| Subject 3 | 35 | 40 | Querétaro | 9.1 Km |
| Subject 4 | 26 | 43 | Amealco de Bonfil | 71.1 Km |
| Subject 5 | 43 | 32 | Querétaro | 9.4 Km |
| Subject 6 | 18 | 34 | San Luis Potosí | 215 Km |
| Subject 7 | 28 | 36 | Querétaro | 10.7 km |
| Subject 8 | 31 | 38 | Querétaro | 18.3 Km |
| Subject 9 | 19 | 43 | Colima | 576 Km |
| Subject 10 | 21 | 47 | Querétaro | 13.5 Km |
| Subject 11 | 39 | 51 | Querétaro | 30.4 Km |
| Subject 12 | 42 | 53 | Querétaro | 13.1 Km |
| Subject 13 | 29 | 54 | El Marqués | 7.1 Km |
| Subject 14 | 33 | 57 | El Marqués | 6.8 Km |
| Subject 15 | 36 | 58 | Salamanca | 89.7 Km |
| Subject 16 | 25 | 58 | Puebla | 329 Km |
| Subject 17 | 42 | 58 | Guadalajara | 378 Km |
| Subject 18 | 45 | 61 | Querétaro | 2.1 Km |
| Subject 19 | 38 | 75 | Querétaro | 17.4 Km |

Experimental tests were conducted in the State of Querétaro. In the first stage, health care professionals of a Women and Child hospital in the city of Querétaro assessed patients. Patients were trained to handle the equipment at home, to step by step visualize data on the fetal monitor screen, and even to place two sensors on the abdominal region of the patient. One sensor measures the fetal heart rate, and the other measures the uterine activity of the patient. In the second stage, experimental tests for the fetal monitor were conducted with patients from remote sites. Data generated by each patient were sent to an OMC in the city of Querétaro. The locations of patients transmitting their records were in many different areas of the State of Queretaro, or in places nearby the State.

Distances from these sites to the OMC were 180 Km on average. Nineteen patients were evaluated, with around 40 studies for each patient. The number of measurements for each clinical study was 900, on average, to prove the approach proposed in this work, providing measurement reliability for fetal heart rate and uterine activity. Figure 7 shows some images of the sites where tests were conducted.

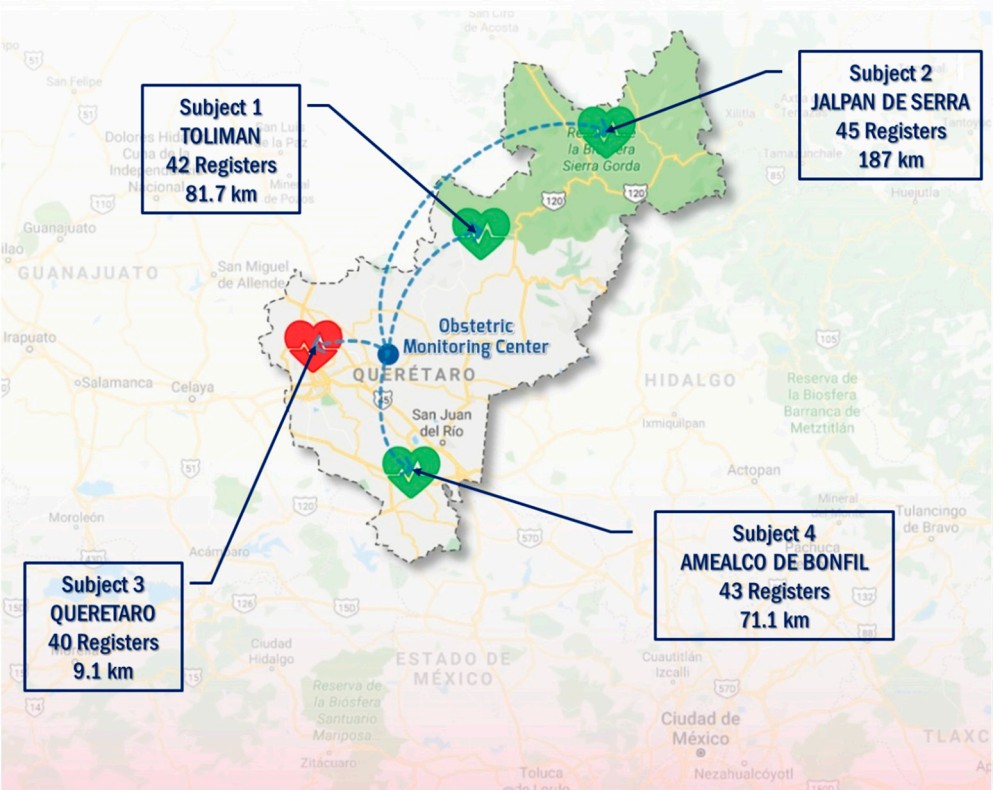

**Figure 7.** Map showing the location of the first four patients from Table 1.

Samples for each medical device were received and stored at OMC. The objective at this stage was to generate a database of unprocessed signals, containing bioelectrical perturbations, to implement and to validate the proposed methodology. Only fetal heart rate signals were processed.

## 4. Results and Discussion

Data from clinical studies carried out on 19 patients were used for system modeling and validating the proposed approach. The clinical study data was divided into two subsets, the first to model the system (data for modeling) and the second to validate the proposed approach (data for validation). The proposed approach was implemented in MATLAB® for building the model, calculating the compensated uncertainty, and evaluating the reliability.

### 4.1. Fetal Monitor Model

To obtain the dynamic model of the fetal monitor, a calibrated signal generator was used to generate an input signal $u_k$ for obtaining an output $y_k$ from the fetal monitor device used in this study. With these data and applying the least-squares method described in Section 3.1.1, a second-order linear deterministic model $l = 2$ was estimated, the ARX model parameters were $\hat{\theta} = [1.1381, -0.2829, 0.0985, -0447, 0.0907]$. The dynamic model response of the fetal monitor is shown in Figure 8.

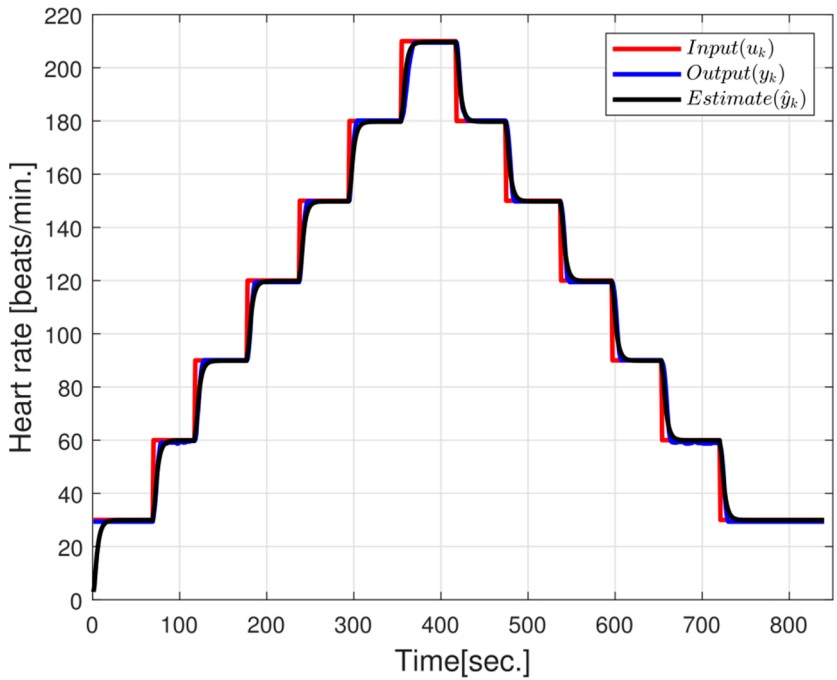

**Figure 8.** Fetal monitor model simulation results.

### 4.2. Stochastic Model

A probabilistic model for each patient was calculated from the experimental data set (data for modeling) acquired from clinical studies carried out on the 19 patients under study. From the fetal heart rate measurements, the perturbations $\xi_k$ was obtained, using the dynamic model of the fetal monitor applying Equation (18), for each patient. Data for ten clinical studies per patient were used to obtain the probabilistic model. From perturbations histogram, estimations of probability density functions $f_{\Xi_n}$ and distribution functions $F_{\Xi_n}$ were obtained, finally a nonparametric model of the inverse function $F_{\Xi_n}^{-1}$ was available for each patient. The direct and inverse distribution functions were estimated by using a linear interpolation by parts [26]. Using the Monte-Carlo method with a large data number of the amount of $N = 10^5$ and $K = 10^3$, the probabilistic models $F_{\Xi_n}^{-1}$ were simulated applying the procedure described in Figure 4 and $s_K^2$ was estimated to calculate the variance using $\sigma^2 = E\{s_K^2\}$. The probability distribution function models of the patients used in the simulation are shown in Figure 9, and Table 2 shows the estimated standard deviations and data number $n$ used in the $F_{\Xi_n}$ modeling.

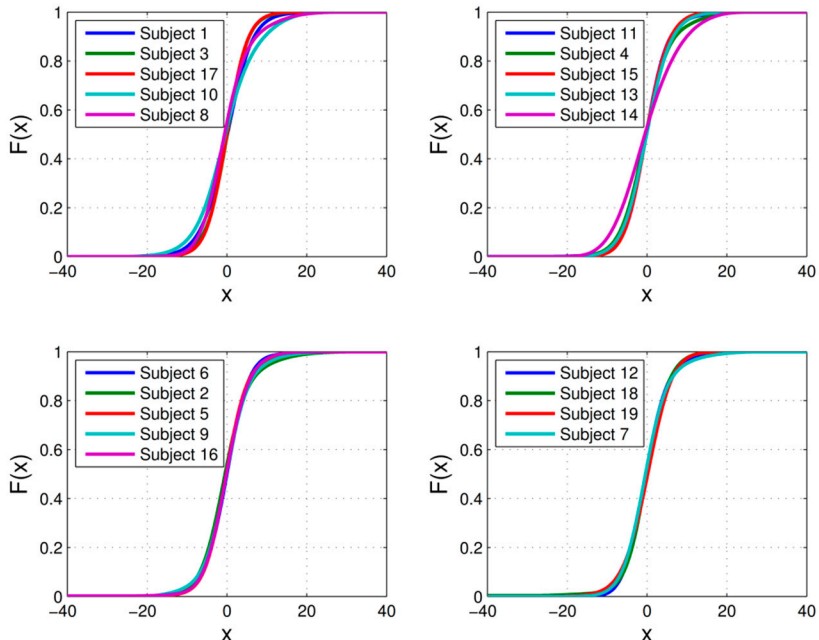

**Figure 9.** Modeled distribution functions of the patients.

**Table 2.** Estimated variance and data number used in stochastic modeling.

| Patient Name | $\sigma$ | $n$ |
|---|---|---|
| Subject 1 | 7.75 | 12731 |
| Subject 2 | 10.98 | 13821 |
| Subject 3 | 12.77 | 9882 |
| Subject 4 | 13.97 | 12062 |
| Subject 5 | 9.19 | 13472 |
| Subject 6 | 8.54 | 12778 |
| Subject 7 | 10.52 | 13417 |
| Subject 8 | 9.44 | 13125 |
| Subject 9 | 11.13 | 12156 |
| Subject 10 | 12.75 | 10243 |
| Subject 11 | 8.64 | 13454 |
| Subject 12 | 8.68 | 13357 |
| Subject 13 | 9.61 | 13399 |
| Subject 14 | 12.97 | 12410 |
| Subject 15 | 6.88 | 13316 |
| Subject 16 | 7.70 | 11677 |
| Subject 17 | 7.49 | 13481 |
| Subject 18 | 11.93 | 12010 |
| Subject 19 | 8.07 | 14070 |

Finally, 100 variance correction coefficients were calculated for each patient, using the relationship defined in Equation (25).

### 4.3. Stochastic Model Validation

From the experimental data set (data for validation), the correction coefficients $c_n$ (calculated during the modeling process) and the algorithm proposed in Figure 5, the corrected standard deviation and the expected fetal heart rate were estimated. For the validation process, $N = 5$ clinical studies with $M = 900$ samples per clinical study and $\beta$ defined with $k = 1$ were considered to determine the minimum percentage of samples that fall within one standard deviation. Table 3 presents the results of

the modeled standard deviation $\sigma$, estimated $s_n$, and corrected $c_n s_n$, and the expected value and $\overline{y}_n$ with samples from five clinical studies for all patients.

**Table 3.** Expected Value and Standard Deviation Estimation of Fetal Heart Rate.

| Patient Name | $\sigma$ | $\overline{y}_5$ | $s_2$ | $c_2 s_2$ | $s_3$ | $c_3 s_3$ | $s_4$ | $c_4 s_4$ | $s_5$ | $c_5 s_5$ |
|---|---|---|---|---|---|---|---|---|---|---|
| Subject 1 | 7.75 | 148.6 | 3.70 | 7.29 | 4.51 | 7.65 | 4.85 | 7.54 | 5.15 | 7.52 |
| Subject 2 | 10.98 | 142.9 | 5.10 | 10.23 | 6.44 | 10.98 | 7.23 | 11.27 | 7.75 | 11.37 |
| Subject 3 | 12.77 | 157.9 | 6.23 | 12.45 | 7.75 | 13.24 | 8.67 | 13.49 | 9.14 | 13.36 |
| Subject 4 | 13.97 | 140.7 | 6.77 | 13.37 | 8.31 | 14.12 | 9.41 | 14.61 | 9.89 | 14.48 |
| Subject 5 | 9.19 | 139.4 | 4.36 | 8.68 | 5.48 | 9.32 | 6.02 | 9.38 | 6.35 | 9.32 |
| Subject 6 | 8.54 | 140.6 | 3.99 | 7.95 | 4.99 | 8.51 | 5.50 | 8.57 | 6.05 | 8.86 |
| Subject 7 | 10.52 | 131.6 | 5.08 | 10.17 | 6.31 | 10.77 | 7.02 | 10.93 | 7.38 | 10.82 |
| Subject 8 | 9.44 | 138.3 | 4.48 | 8.92 | 5.57 | 9.49 | 6.31 | 9.83 | 6.68 | 9.80 |
| Subject 9 | 11.13 | 139.5 | 5.25 | 10.37 | 6.65 | 11.27 | 7.29 | 11.26 | 7.89 | 11.43 |
| Subject 10 | 12.75 | 132.9 | 6.26 | 12.67 | 7.78 | 13.32 | 8.68 | 13.56 | 9.30 | 13.63 |
| Subject 11 | 8.64 | 123.7 | 4.05 | 8.03 | 5.17 | 8.77 | 5.68 | 8.84 | 6.17 | 9.02 |
| Subject 12 | 8.68 | 135.7 | 3.96 | 7.92 | 5.12 | 8.70 | 5.70 | 8.87 | 6.07 | 8.87 |
| Subject 13 | 9.61 | 148.0 | 4.51 | 9.03 | 5.67 | 9.65 | 6.40 | 9.93 | 6.82 | 9.95 |
| Subject 14 | 12.97 | 141.3 | 6.39 | 12.72 | 8.10 | 13.83 | 8.96 | 13.95 | 9.50 | 13.92 |
| Subject 15 | 6.88 | 129.7 | 3.14 | 6.22 | 3.92 | 6.68 | 4.36 | 6.80 | 4.58 | 6.70 |
| Subject 16 | 7.70 | 134.2 | 3.52 | 7.07 | 4.42 | 7.55 | 4.99 | 7.76 | 5.38 | 7.86 |
| Subject 17 | 7.49 | 141.5 | 3.40 | 6.82 | 4.27 | 7.30 | 4.72 | 7.34 | 5.03 | 7.35 |
| Subject 18 | 11.93 | 132.8 | 6.04 | 11.84 | 7.37 | 12.49 | 8.11 | 12.58 | 8.59 | 12.51 |
| Subject 19 | 8.07 | 150.6 | 3.66 | 7.29 | 4.67 | 7.96 | 5.29 | 8.26 | 5.62 | 8.25 |

The algorithm in Figure 5, additionally, allows the evaluation of the reliability improvement using the corrected and uncorrected standard deviation (uncertainty). To verify the reliability behavior, the proposed algorithm was executed using two bands level: $\beta = k^2 c_n$ to obtain the corrected reliability value and $\beta = k^2$ to obtain the reliability value without correction. Figure 10 shows the results of the corrected and uncorrected reliability for two clinical studies $n = \sqrt{2}$, Figure 10a, and for five clinical studies $n = 5$, Figure 10b. The reliability increases when considering the corrected standard deviation (uncertainty).

In the case of total ignorance of the system, the Chebyshev inequality indicates that at least $1 - \frac{1}{k^2}$ of the sample, data must be within $k$ standard deviations of the mean, for $k = \sqrt{2}$ for at least 50% of the data, see Figure 10. However, having the distribution function models allow us, in addition to estimating the compensated uncertainty, to know the number of measurements that were within $\sqrt{2}$ standard deviations, which was close to 90%.

The parameter estimator behavior in traditional models [12–14], was performed considering some theoretical distributions [3], and their behavior was evaluated considering finite samples. The problems that arose were that the distributions used were not representative of the applications considered in this work, and the estimator properties within finite time windows depended on the distributions. The method proposed in this work allows defining distributions in practical applications with few samples; that solve the problems presented in classical models.

Reliability determination with few samples could be useful due to the visual analysis of fetal heart rate tracings being subjective and inconsistent [5]. There is disagreement in the clinical decision in fetal heart rate (FHR) classification [7]. A simpler and more objective set of guidelines could provide better reliability [10]. Given the inherent limitations of electronic fetal monitoring technology and of the ability of human beings to characterize fetal heart rate parameters and patterns, the low-reliability degree may represent the best-case scenario of the visual fetal heart rate interpretation reliability [6].

In the international federation of gynecology and obstetrics (FIGO), national institute for health and care (NICE), and American college of obstetrics and gynecology (ACOG) guidelines [8], there are two fundamental parameters to determine normal, suspicious, and pathological patterns. The first is

the heart rate baseline, and the other is variability. The method proposed in this work can be applied directly to a correct estimate of the variability because it includes the specific distribution function of each patient. In the tests analyzed, differences of 10% were found in the estimate (n = 5). According to results, the baseline is not affected by the distribution.

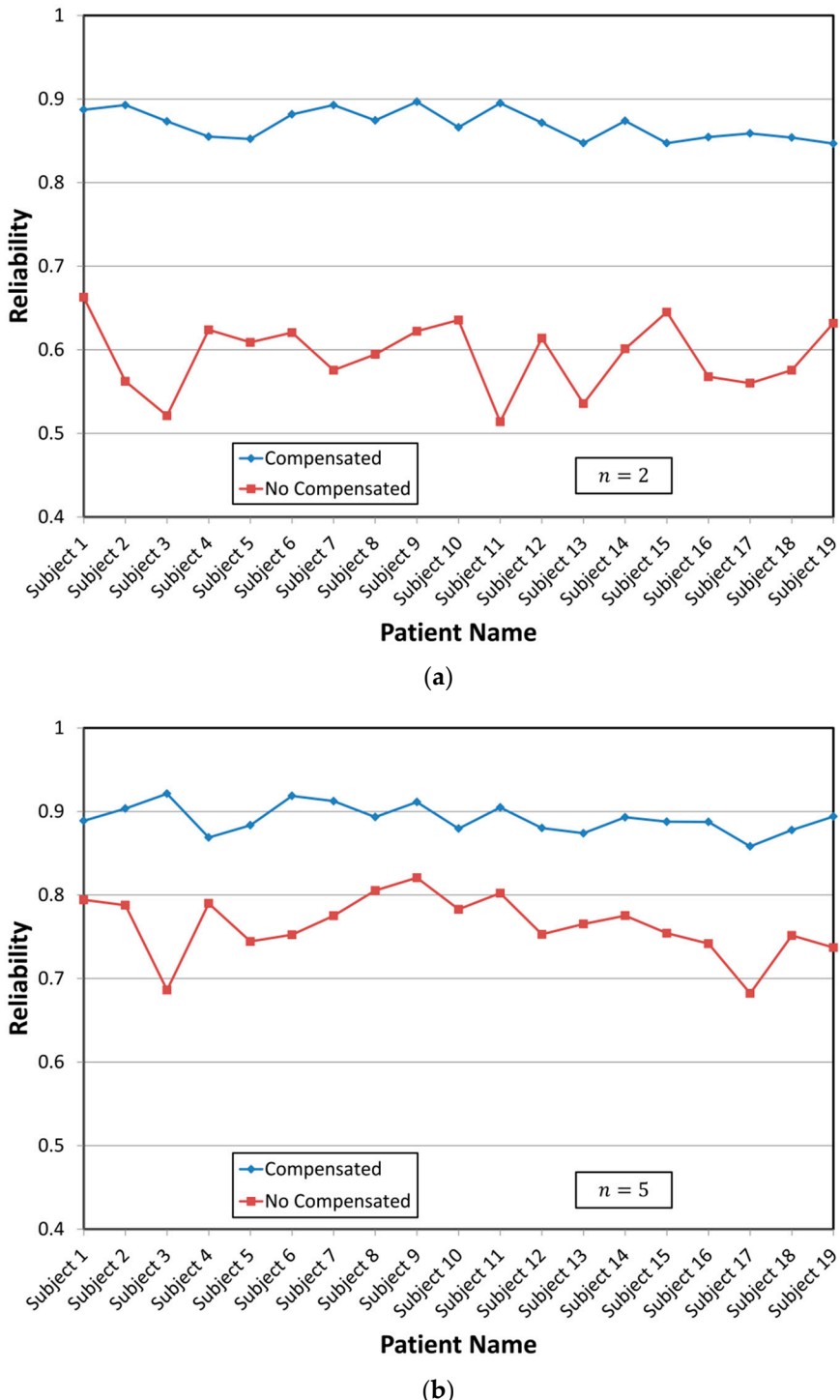

**Figure 10.** Corrected and uncorrected reliability (R) calculations; (**a**) with two clinical studies; (**b**) with five clinical studies.

## 5. Conclusions

Reliability has reached an important level for consideration at the beginning of product design, specifically for medical applications, which have been favored by including, as a quality criterion, the reliability level of the device.

Effect analysis of the distribution function and sample lengths in the typical estimators' calculation, mean, and standard deviation were presented. Specific distribution estimates of each patient were used; this procedure suggests a natural way to integrate the additional information obtained in each study. Several studies carried out on patients verified that the real distributions were far from the theoretical distributions that are usually used to determine the estimators' properties. Furthermore, these distributions were used to determine the factors necessary to obtain an unbiased estimator of uncertainty. This information was used to implement a simple algorithm that improves reliability estimation. With the available calculation tool, it is possible to obtain specific information that allows improving the estimators' behavior used in decision making without assuming hypotheses that are difficult to verify in daily applications. Applying classical theory without verifying that hypotheses are met may lead to incorrect results. However, it was demonstrated that these might be corrected by using compensation factors.

Considering how fast available electronic systems for the integration of medical devices change, the way reliability is assessed, restrictions imposed by classical estimation theory, and computer resources, the developed methodology attempts to be the spearhead of a series of works that would lead to a more accurate estimation of reliability.

**Author Contributions:** Conceptualization, S.A.-M. and R.S.-Z.; methodology, G.R.-L. and S.A.-M.; software, L.B.-R.; validation, G.R.-L., S.A.-M. and R.S.-Z.; formal analysis, R.S.-Z.; investigation, T.S.-J.; resources, S.A.-M.; data curation, R.S.-Z. and L.B.-R.; writing—original draft preparation, G.R.-L.; writing—review and editing, S.A.-M. and T.S.-J.; visualization, G.R.-L.; supervision, T.S.-J.; project administration, S.A.-M.; funding acquisition, S.A.-M. All authors have read and agreed to the published version of the manuscript.

**Funding:** This work was supported by the FOMIX, Queretaro State Mexico mixed foundation (Grant: QRO-2010-C02-149261).

**Acknowledgments:** Authors acknowledge Carlos Arturo Rebolledo Fernandez and Julio Cesar Ramirez Arguello for supporting the project logistic.

**Conflicts of Interest:** The authors declare no conflict of interest.

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
