# Peer review of "Compensation of Measurement Uncertainty in a Remote Fetal Monitor"

_applsci, doi:10.3390/app10093274_

Round 1
Reviewer 1 Report
The authors have properly addressed my comments.
Author Response
Querétaro Qro., México, April 23th, 2020
Dear Prof. Dr. Takayoshi Kobayashi
Editor-in-Chief
MDPI applied sciences-Journal
Advanced Ultrafast Laser Research Center, The University of Electro-Communications.
Dear Dr. Takayoshi Kobayashi.
All comments of the reviewers were taken into account. Suggestions and comments of the Reviewer 1 describing the changes made to the manuscript are attached about our contribution titled “Compensation of Measurement Uncertainty in a Remote Fetal Monitor” submitted for the Journal MDPI applied sciences (applsci-785710); I hope these changes had been enough.
Sincerely Yours,
Dr. Guillermo Ronquillo Lomeli
Professor
Engineering and Industrial Development Center
702 Playa pie de la cuesta
Querétaro, Qro. México 76125. +52 (442) 211-9800
+52 (442) 211-9800.
email: gronquillo@cidesi.edu.mx

Reviewer 2 Report
This version of the paper is much better than the previous one. You have addressed all my remarks from previous review and I have no more negative comments. I still feel that the paper is better suited for other journal dealing with medicine or applied statistics but for sure it gives an example of applying science to real life problems so I recommend it for publication.
Author Response
Querétaro Qro., México, April 23th, 2020
Dear Prof. Dr. Takayoshi Kobayashi
Editor-in-Chief
MDPI applied sciences-Journal
Advanced Ultrafast Laser Research Center, The University of Electro-Communications.
Dear Dr. Takayoshi Kobayashi.
All comments of the reviewers were taken into account. Suggestions and comments of the Reviewer 2 describing the changes made to the manuscript are attached about our contribution titled “Compensation of Measurement Uncertainty in a Remote Fetal Monitor” submitted for the Journal MDPI applied sciences (applsci-785710); I hope these changes had been enough.
Sincerely Yours,
Dr. Guillermo Ronquillo Lomeli
Professor
Engineering and Industrial Development Center
702 Playa pie de la cuesta
Querétaro, Qro. México 76125. +52 (442) 211-9800
+52 (442) 211-9800.
email: gronquillo@cidesi.edu.mx.
